# Influence of vitamin D supplementation on growth, body composition, pubertal development and spirometry in South African schoolchildren: a randomised controlled trial (ViDiKids)

Keren Middelkoop,[1,2] Lisa Micklesfield,[3,4] Justine Stewart,[1,2] Neil Walker,[5] David A Jolliffe,[6] Amy E Mendham,[3,4] Anna K Coussens,[7,8] James Nuttall,[9] Jonathan Tang,[10,11] William D Fraser,[10,11] Waheedullah Momand,[6] Cyrus Cooper,[12,13] Nicholas C Harvey,[12,13] Robert J Wilkinson,[7,14,15] Linda-Gail Bekker,[1,2] Adrian R Martineau ![ORCID][6]

KM and LM contributed equally.

KM and LM are joint first authors.

For numbered affiliations see end of article.

**Correspondence to**
Prof Adrian R Martineau; a.martineau@qmul.ac.uk

## ABSTRACT

**Objective** To determine whether weekly oral vitamin D supplementation influences growth, body composition, pubertal development or spirometric outcomes in South African schoolchildren.

**Design** Phase 3 double-blind randomised placebo-controlled trial.

**Setting** Socioeconomically disadvantaged peri-urban district of Cape Town, South Africa.

**Participants** 1682 children of black African ancestry attending government primary schools and aged 6–11 years at baseline.

**Interventions** Oral vitamin $D_3$ (10 000 IU/week) versus placebo for 3 years.

**Main outcome measures** Height-for-age and body mass index-for-age, measured in all participants; Tanner scores for pubertal development, spirometric lung volumes and body composition, measured in a subset of 450 children who additionally took part in a nested substudy.

**Results** Mean serum 25-hydroxyvitamin $D_3$ concentration at 3-year follow-up was higher among children randomised to receive vitamin D versus placebo (104.3 vs 64.7 nmol/L, respectively; mean difference (MD) 39.7 nmol/L, 95% CI 37.6 to 41.9 nmol/L). No statistically significant differences in height-for-age z-score (adjusted MD (aMD) −0.08, 95% CI −0.19 to 0.03) or body mass index-for-age z-score (aMD −0.04, 95% CI −0.16 to 0.07) were seen between vitamin D versus placebo groups at follow-up. Among substudy participants, allocation to vitamin D versus placebo did not influence pubertal development scores, % predicted forced expiratory volume in 1 s (FEV1), % predicted forced vital capacity (FVC), % predicted FEV1/FVC, fat mass or fat-free mass.

**Conclusions** Weekly oral administration of 10 000 IU vitamin $D_3$ boosted vitamin D status but did not influence growth, body composition, pubertal development or spirometric outcomes in South African schoolchildren.

**Trial registration numbers** ClinicalTrials.gov NCT02880982, South African National Clinical Trials Register DOH-27-0916-5527.

## WHAT IS ALREADY KNOWN ON THIS TOPIC

⇒ Observational studies have reported independent associations between vitamin D deficiency in childhood and slower linear growth, reduced lean mass, obesity and precocious puberty.

⇒ A phase 2 clinical trial conducted in Mongolia reported that a 6-month course of vitamin D supplementation increased height gain in 113 vitamin D deficient schoolchildren aged 12–15 years; however, these results were not confirmed by a recent phase 3 trial conducted in the same setting.

⇒ Randomised controlled trials to determine the effects of vitamin D supplementation on growth and development in schoolchildren have not been conducted in other settings.

## WHAT THIS STUDY ADDS

⇒ This placebo-controlled phase 3 clinical trial, conducted in 1682 black African schoolchildren in Cape Town, South Africa, showed that a 3-year course of weekly vitamin D supplementation was effective in elevating circulating 25-hydroxyvitamin D concentrations.

⇒ However, this was not associated with any effect on linear growth, body composition, pubertal development or spirometric lung volumes.

## HOW THIS STUDY MIGHT AFFECT RESEARCH, PRACTICE OR POLICY

⇒ Our findings do not support the use of vitamin D supplementation as an intervention to influence child growth, body composition, pubertal development or spirometric lung volumes.

## INTRODUCTION

Middle childhood and early adolescence represent key periods for growth and development that have an important influence

on stature and health outcomes in later adolescence and adulthood.[1] Stunting in childhood has long been recognised to associate with multiple adverse long-term health outcomes,[2] and is particularly common in lower-income countries.[3] These settings have also witnessed an emerging epidemic of childhood obesity,[4] which has in turn been associated with accelerated pubertal development.[5] Interventions to alleviate these public health challenges are urgently needed.

Vitamin D is a fat-soluble micronutrient with pleotropic effects on human health.[6] Observational studies have reported that vitamin D deficiency, as evidenced by low circulating concentrations of its major metabolite 25-hydroxyvitamin D (25(OH)D), associates with slower linear growth,[7] reduced lean mass,[8] childhood obesity[9–11] and precocious puberty,[12 13] potentially reflecting the ability of vitamin D to stimulate production of insulin-like growth factor 1[14] and regulate adipogenesis.[15] A phase 2 randomised controlled trial (RCT) conducted in Mongolia reported that a 6-month course of vitamin D supplementation increased height gain in 113 vitamin D deficient schoolchildren aged 12–15 years,[16] but these results were not confirmed by a recent phase 3 RCT conducted in the same setting.[17] RCTs to determine the effects of vitamin D supplementation on growth and development in schoolchildren have not been conducted in other settings representing children at different risks of malnutrition, stunting and obesity. An opportunity to conduct such an investigation recently arose as part of the ViDiKids trial, a multicentre phase 3 RCT that investigated the effects of weekly oral administration of 10 000 IU vitamin $D_3$ for 3 years on the primary outcome of tuberculosis infection in a cohort of 1682 schoolchildren aged 6–11 years living in a socioeconomically disadvantaged peri-urban district of Cape Town, South Africa.[18] Height-for-age z-scores and body mass index (BMI)-for-age z-scores were assessed in all participants (n=1682), and Tanner scores for pubertal development, spirometric lung volumes and body composition were assessed in a subset of 450 children who also took part in a nested substudy.

## METHODS

### Trial design, setting and sponsorship

We conducted a multicentre phase 3 double-blind individually randomised placebo-controlled trial of weekly oral vitamin D supplementation in 23 government schools in Cape Town, South Africa, as previously described.[18 19] The primary outcome was the acquisition of tuberculosis infection, as evidenced by the conversion of a QuantiFERON-TB Gold Plus (QFT-Plus) assay result from negative at baseline to positive at 3-year follow-up. The current manuscript reports the effects of the intervention on prespecified secondary outcomes relating to growth in all study participants, and body composition, pubertal development and spirometry in a subset of participants who

additionally took part in a nested substudy. The trial was sponsored by Queen Mary University of London.

### Participants

Inclusion criteria for the main trial were enrolment in Grades 1–4 at a participating school; age 6–11 years at screening; and written informed assent/consent to participate in the main trial provided by children and their parent/legal guardian, respectively. Exclusion criteria for the main trial were a history of previous tuberculosis infection, active tuberculosis disease or any chronic illness other than asthma (including known or suspected HIV infection) prior to enrolment; use of any regular medication other than asthma medication; use of vitamin D supplements at a dose of more than 400 IU/day in the month before enrolment; plans to move away from study area within 3 years of enrolment; inability to swallow a placebo soft gel capsule with ease; and clinical evidence of rickets or a positive QFT-Plus assay result at screening. An additional inclusion criterion for the substudy was enrolment in Grade 4 at a participating school (ie, only children in Grade 4 were eligible for the substudy).

### Enrolment

Full details of enrolment procedures are described in Supplementary Methods (online supplemental material). Eligible participants underwent measurement of weight using a Digital Floor Scale (Charder Medical, Taichung City, Taiwan), height using a portable stadiometer (HM200P, Charder Medical) and waist circumference using a measuring tape. Substudy participants also underwent spirometry according to European Respiratory Society and American Thoracic Society standards[20] using a portable spirometer (Carefusion, San Diego, California, USA) and measurement of body composition by dual-energy X-ray absorptiometry (DXA) as described below.

### Randomisation and blinding

Full details of randomisation and blinding procedures have been described previously,[18 19] and are presented in Supplementary Methods (online supplemental material). Briefly, eligible and assenting children whose parents consented to their participation in the trial were individually randomised to receive a weekly capsule containing vitamin $D_3$ or placebo for 3 years, with a one-to-one allocation ratio and randomisation stratified by the school of attendance. Treatment allocation was concealed from participants, care providers and all trial staff (including senior investigators and those assessing outcomes) until completion of the trial to maintain the double-blind.

### Intervention

Study medication comprised a 3-year course of weekly soft gel capsules manufactured by the Tishcon Corporation (Westbury, New York, USA), containing either 0.25 mg (10 000 international units) cholecalciferol (vitamin $D_3$) in olive oil (intervention arm) or olive oil without any vitamin $D_3$ content (placebo arm). A weekly dose of

10 000 IU (equivalent to 1429 IU/day) was selected in preference to a daily dose of 600 IU (the Recommended Daily Allowance for this age group) because we were concerned that the latter dose would be inadequate to maintain serum 25(OH)D concentrations >50 nmol/L,[21] and we felt that adherence to a directly-observed weekly dose would be superior to daily self-administration.[22] Weekly supplementation has also been shown to be effective in elevating 25(OH)D concentrations into the physiological range in children by other investigators.[23] Active and placebo capsules had identical appearance and taste. Capsules were taken under direct observation of study staff during school term time. Further details of the administration of study medication are provided in Supplementary Methods (online supplemental material).

### Follow-up assessments

At 1-year, 2-year and 3-year follow-up, height, weight and waist circumference were measured as at baseline. At 3-year follow-up, substudy participants were additionally invited to undergo repeat spirometry and DXA scanning and to complete a Tanner self-assessment questionnaire for pubertal development.[24]

### Outcomes

The primary outcome of the main trial, reported elsewhere,[18] was the QFT-Plus result at the end of the study. The following secondary outcomes were assessed for all participants: height-for-age, BMI-for-age, waist circumference-for-age and waist-to-height ratio (all participants). Additional outcomes assessed in substudy participants only were: whole body fat mass, fat-free soft tissue mass, % predicted forced vital capacity (FVC), % predicted forced expiratory volume in 1 s (FEV1), % predicted FEV1/FVC, mean Tanner scores for pubic hair (men and women), the external genitalia (men only) or breast development (women only), the proportion of participants reaching menarche by the end of the trial (women only) and mean age at menarche (women who reached menarche by the end of the trial only).

### DXA

Body composition (whole body fat mass and fat-free soft tissue mass) was assessed using a Hologic Discovery-W DXA scanner at the Sports Science Institute of South Africa, University of Cape Town. All scans were performed by a trained radiographer on one scanner (Hologic, Bedford, Massachusetts, USA) using standard procedures, and analysed using Apex software (V.13.4.1). Quality assurance checks were carried out prior to scanning and generated coefficients of variation <0.5%. DXA outcomes relating to the effects of the trial intervention on bone mineral content have been reported elsewhere.[19]

### Laboratory assessments

Serum concentrations of 25(OH)D$_3$ were measured using liquid chromatography-tandem mass spectrometry as previously described.[25] Further details are presented in Supplementary Methods (online supplemental material).

### Sample size

The sample size for the main trial was calculated as the number needed to detect a 25% reduction in the proportion of children with a positive QFT-Plus assay result at a 3-year follow-up (primary outcome) with 80% power and 5% type 1 error, assuming a 3.5% annual risk of QFT-Plus conversion, 20% loss to follow-up and a 5% risk of an indeterminate QFT-Plus assay result at the end of the study.[18] The sample size for the substudy was calculated as the number needed to detect an inter-arm difference of 0.35 SDs in mean bone mineral content at the whole body less head and lumbar spine with 88% power and 5% type 1 error, assuming 29% loss to follow-up at 3 years.[19]

### Statistical analyses

Full details of statistical analyses are provided in Supplementary Methods (online supplemental material). Briefly, effects of treatment on age-adjusted and sex-adjusted z-scores for anthropometric outcomes were estimated by fitting allocation to vitamin D versus placebo as the sole fixed effect in a mixed effects linear regression model with a random effect for repeated assessments of each individual participant and a random effect of the school of attendance. Effects of treatment on DXA and spirometric outcomes were analysed using multilevel mixed models with adjustment for baseline values and a random effect for school. Tanner metrics were analysed in a similar fashion, but restricted to the applicable sex and without baseline adjustment. Prespecified subgroup analyses were conducted to determine whether the effect of vitamin D supplementation was modified by sex (male vs female), baseline deseasonalised 25(OH)D$_3$ concentration (<75 vs ≥75 nmol/L)[26] and calcium intake (< vs ≥ median value of 466 mg/day).[19] These were performed by repeating efficacy analyses with the inclusion of an interaction term between allocation (to vitamin D vs placebo) and each posited effect-modifier with the presentation of the p value associated with this interaction term. An Independent Data Monitoring Committee reviewed accumulating serious adverse event data at 6-monthly intervals, and recommended continuation of the trial at each review. No interim efficacy analysis was performed.

### Patient and public involvement

The Desmond Tutu HIV Centre's Community Advisory Group was consulted on the design and conduct of the ViDiKids trial.

## RESULTS

### Participants

A total of 2852 children were screened for eligibility from March 2017 to March 2019, of whom 2271 underwent QFT testing: 1682 (74.1%) QFT-negative children were randomly assigned to receive vitamin D$_3$ (829

participants) or placebo (853 participants) as previously described.[18] 450/1682 (26.8%) participants in the main trial also participated in the substudy, of whom 228 versus 222 participants were allocated to the vitamin D versus placebo arms, respectively (figure 1). Table 1 presents baseline characteristics of children in the main trial and

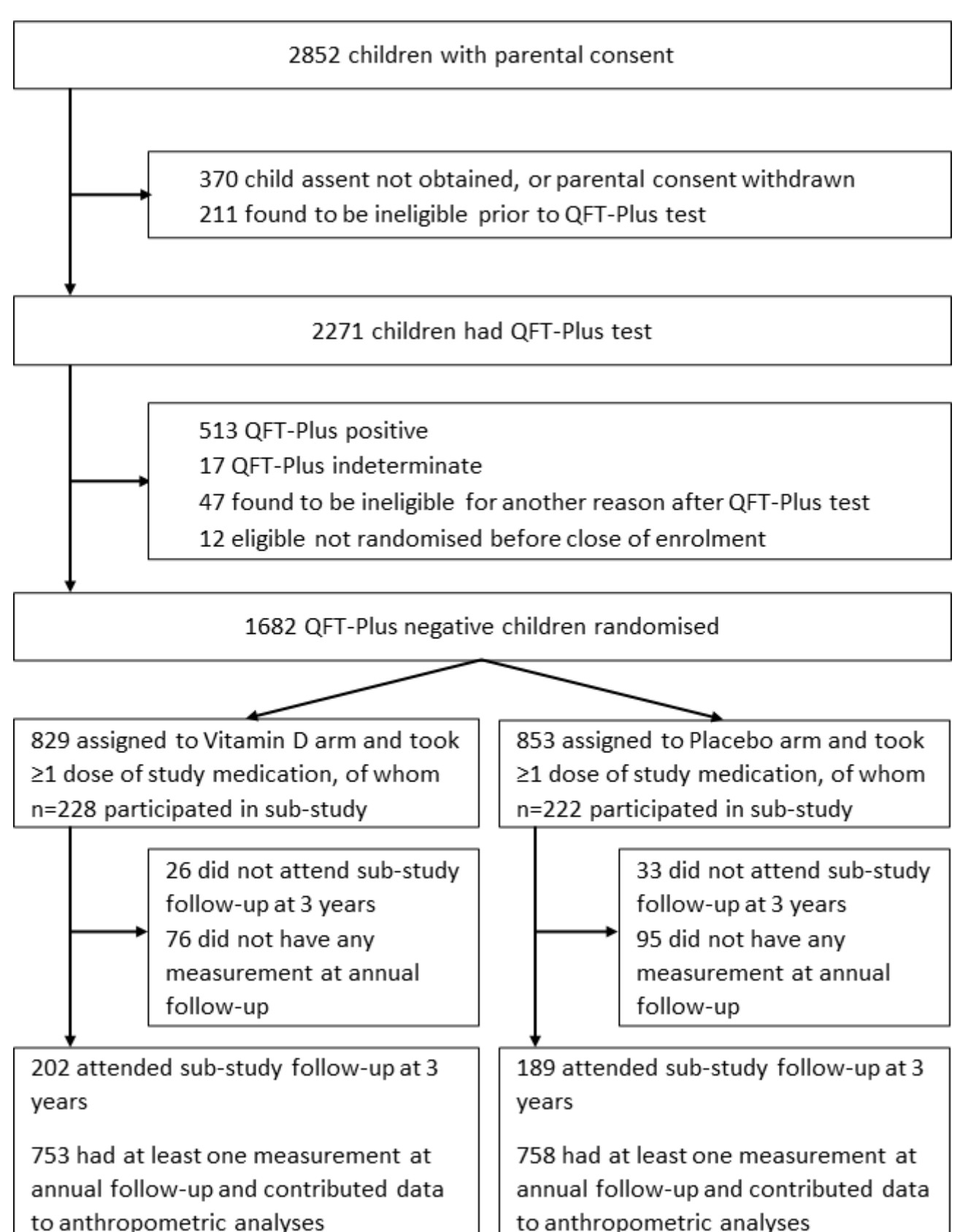

**Figure 1** Participant flow diagram. QFT, QuantiFERON-TB.

**Table 1** Participants' baseline characteristics by allocation: main study and substudy

| | | Main study (n=1682) | | | Substudy (n=450 subset) | | |
|---|---|---|---|---|---|---|---|
| | | Overall (n=1682) | Vitamin D arm (n=829) | Placebo arm (n=853) | Overall (n=450) | Vitamin D arm (n=228) | Placebo arm (n=222) |
| Mean age, years (SD) | | 8.9 (1.4) | 8.9 (1.4) | 8.8 (1.3) | 10.1 (0.7) | 10.2 (0.7) | 10.0 (0.6) |
| Female sex, n (%) | | 880 (52.4) | 437 (52.8) | 443 (51.9) | 234 (52.0) | 116 (50.9) | 118 (53.2) |
| Ethnic origin* | Xhosa, n (%) | 1615 (97.9) | 788 (97.3) | 827 (98.5) | 424 (96.4) | 214 (96.0) | 210 (96.8) |
| | Other, n (%) | 35 (2.1) | 22 (2.7) | 13 (1.5) | 16 (3.6) | 9 (4.0) | 7 (3.2) |
| Type of residence | Brick, n (%) | 867 (51.5) | 423 (51.0) | 444 (52.1) | 230 (51.1) | 121 (53.1) | 109 (49.1) |
| | Informal, n(%) | 815 (48.5) | 406 (49.0) | 409 (47.9) | 220 (48.9) | 107 (46.9) | 113 (50.9) |
| Parental education*† | Primary school, n (%) | 60 (3.6) | 34 (4.1) | 26 (3.1) | 24 (5.3) | 16 (7.0) | 8 (3.6) |
| | Secondary school or higher, n (%) | 1618 (96.4) | 792 (95.9) | 826 (96.9) | 426 (94.7) | 212 (93.0) | 214 (96.4) |
| Mean monthly household income, 1000 ZAR (SD) | | 1.88 (2.20) | 1.81 (2.15) | 1.95 (2.25) | 1.57 (2.31) | 1.52 (2.69) | 1.63 (1.86) |
| Mean height-for-age z-score (SD)* | | −0.55 (1.20) | −0.56 (1.26) | −0.54 (1.15) | −0.43 (0.96) | −0.43 (0.98) | −0.44 (0.95) |
| Height-for-age z-score category | <−2.00 (stunted), n (%) | 186 (11.1) | 99 (11.9) | 87 (10.2) | 29 (6.4) | 16 (7.0) | 13 (5.9) |
| | ≥−2.00, n (%) | 1496 (88.9) | 730 (88.1) | 766 (89.8) | 421 (93.6) | 212 (93.0) | 209 (94.1) |
| Mean BMI-for-age z-score (SD)* | | 0.33 (1.03) | 0.31 (1.03) | 0.34 (1.03) | 0.20 (0.97) | 0.19 (0.92) | 0.21 (1.02) |
| BMI-for-age z-score category | <−2.00 (thin), n (%) | 26 (1.5) | 15 (1.8) | 11 (1.3) | 7 (1.6) | 4 (1.8) | 3 (1.4) |
| | >−2.00 and ≤1.00, n (%) | 1223 (72.7) | 602 (72.6) | 621 (72.8) | 354 (78.7) | 178 (78.1) | 176 (79.3) |
| | >1.00 and ≤2.00 (overweight), n (%) | 340 (20.2) | 165 (19.9) | 175 (20.5) | 72 (16.0) | 42 (18.4) | 30 (13.5) |
| | >2.00 (obese), n (%) | 93 (5.5) | 47 (5.7) | 46 (5.4) | 17 (3.8) | 4 (1.8) | 13 (5.9) |
| Mean waist circumference-for-age z-score (SD)* | | −0.13 (0.90) | −0.11 (0.90) | −0.15 (0.91) | −0.20 (0.98) | −0.20 (0.98) | −0.20 (0.98) |
| Mean waist-to-height ratio z-score (SD)* | | 0.03 (0.85) | 0.05 (0.83) | 0.00 (0.86) | −0.05 (0.86) | −0.05 (0.85) | −0.04 (0.88) |
| Mean whole body fat mass, kg (SD) | | – | – | – | 3.71 (2.26) | 3.66 (2.13) | 3.77 (2.40) |
| Mean whole body fat-free soft tissue mass, kg (SD) | | – | – | – | 9.27 (1.95) | 9.39 (2.02) | 9.15 (1.86) |
| % predicted FEV1 (SD) | | – | – | – | 83.2 (15.7) | 83.8 (15.2) | 82.7 (16.2) |

Continued

**Table 1** Continued

| | | Main study (n=1682) | | | Substudy (n=450 subset) | | |
|---|---|---|---|---|---|---|---|
| % predicted FVC (SD) | | – | – | – | 94.8 (18.7) | 94.6 (18.2) | 95.0 (19.3) |
| % predicted FEV1/FVC (SD) | | – | – | – | 91.4 (16.9) | 92.2 (16.3) | 90.5 (17.5) |
| Calcium intake* | <median, n (%)‡ | 817 (50.0) | 383 (47.6) | 434 (52.3) | 248 (56.2) | 119 (53.6) | 129 (58.9) |
| | ≥median, n (%)‡ | 817 (50.0) | 421 (52.4) | 396 (47.7) | 193 (43.8) | 103 (46.4) | 90 (41.1) |
| Mean serum $25(OH)D_3$ concentration, nmol/L (SD)*§ | | 71.2 (14.8) | 71.2 (14.5) | 71.1 (15.0) | 70.0 (13.5) | 70.4 (12.1) | 69.6 (14.9) |
| Serum $25(OH)D_3$ concentration, category*§ | <25 nmol/L, n (%) | 1 (0.1) | 0 (0.0) | 1 (0.1) | 1 (0.3) | 0 (0) | 1 (0.6) |
| | ≥25 nmol/L and <50 nmol/L, n (%) | 74 (5.4) | 34 (5.1) | 40 (5.8) | 20 (5.5) | 7 (3.7) | 13 (7.4) |
| | ≥50 nmol/L and <75 nmol/L, n (%) | 787 (57.7) | 394 (58.8) | 393 (56.6) | 214 (59.0) | 114 (61.0) | 100 (56.8) |
| | ≥75 nmol/L, n (%) | 502 (36.8) | 242 (36.1) | 260 (37.5) | 128 (35.3) | 66 (35.3) | 62 (35.2) |

*Missing data (for bone substudy: ethnicity, n=5 vitamin D arm, n=5 placebo arm; for calcium intake, n=6 vitamin D arm, n=3 placebo arm; for serum 25(OH)D3 concentration, n=66 vitamin D arm, n=62 placebo arm; for serum adjusted calcium and total ALP concentration, n=41 vitamin D arm, n=45 placebo arm; for PTH: n=42 vitamin D arm, n=46 placebo arm; for CTX: n=41 vitamin D arm, n=46 placebo arm; for P1NP: n=43 vitamin D arm, n=46 placebo arm; for fracture study: ethnicity, n=19 vitamin D arm, n=13 placebo arm; parental education, n=3 vitamin D arm, n=1 placebo arm; BMI-for-age z-score, n=2 vitamin D arm, n=0 placebo arm; height-for-age z-score, n=2 vitamin D arm, n=0 placebo arm; for calcium intake, n=25 Vitamin D arm, n=23 placebo arm; for serum 25(OH)D3 concentration, n=159 vitamin D arm, n=159 placebo arm.
†Highest level of education of at least one parent.
‡Median calcium intake 466 mg/day.
§Deseasonalised values.
ALP, alkaline phosphatase; BMC, bone mineral content; BMI, body mass index; CTX, C-terminal cross-linked telopeptide; N, propeptide; 25(OH)D3, 25-hydroxyvitamin D3; P1NP, serum procollagen type I; PTH, parathyroid hormone; ZAR, South African rand.

in the substudy, overall and by study arm. Mean age was higher among participants in the substudy versus all those in the main trial (10.1 vs 8.9 years, respectively), reflecting the fact that participation in the substudy was restricted to children enrolled in Grade 4. Prevalence of obesity and stunting were lower in the substudy versus the main trial (3.8% vs 5.5%, and 6.4% vs 11.1%, respectively). Baseline characteristics were otherwise well balanced for all participants in the main trial vs those who additionally participated in the substudy: 52.4% vs 52.0% were female and mean serum $25(OH)D_3$ concentrations were 71.2 nmol/L vs 70.0 nmol/L. Within the main trial and the substudy, baseline characteristics of those randomised to vitamin D versus placebo were also well balanced. The median duration of follow-up was 3.16 years (IQR, 2.83–3.38 years) and was not different between the two study arms. For the main trial, mean serum $25(OH)D_3$ concentrations at 3-year follow-up were higher among children randomised to receive vitamin D versus placebo (104.3 vs 64.7 nmol/L, respectively; mean difference 39.7 nmol/L, 95% CI for difference 37.6 to 41.9 nmol/L).

### Growth outcomes

Among participants in the main trial, allocation to vitamin D versus placebo did not significantly influence mean height-for-age z-scores at annual follow-up, either overall or within subgroups defined by sex, calcium intake or baseline $25(OH)D$ concentration (table 2: p values for interaction >0.05). Similarly, no statistically significant effect of the intervention was seen on mean BMI-for-age z-scores (table 3), waist circumference-for-age z-scores (online supplemental table S1) or waist-to-height ratio z-scores (online supplemental table S2), either overall or by subgroup (p values for interaction >0.05). Among substudy participants, allocation to vitamin D versus placebo did not significantly influence fat mass or fat-free soft tissue mass (online supplemental table S3) or % predicted FEV1 or FEV1/FVC (online supplemental table S4), either overall or by subgroup. For the outcome of % predicted FVC, allocation to vitamin D versus placebo did not have a statistically significant effect overall, or within subgroups defined by calcium intake or baseline $25(OH)D$ concentration; however, the p value for interaction associated with the subgroup analysis by sex (p=0.049) raised the possibility that this factor might modify the effect of vitamin D on % predicted FVC (online supplemental table S4).

### Developmental outcomes

Among substudy participants, no statistically significant inter-arm differences were seen in mean Tanner scores for pubic hair (men and women), external genitalia (men only), proportion menstruating (women only), mean age at menarche (women only) or breast development (women only), either overall or within subgroups defined by calcium intake or baseline $25(OH)D$ concentration (table 4).

### Adverse events

Incidence of adverse events by the trial arm has been reported elsewhere.[18] No serious events arising in the trial were adjudged related to the administration of vitamin D or placebo.

### DISCUSSION

We report findings from the first RCT to investigate the effects of vitamin D supplementation on growth and developmental indices in schoolchildren of black African ancestry. Administration of oral vitamin D supplementation at a dose of 10 000 IU per week for 3 years was effective in boosting vitamin D status, but it did not have statistically significant effects on linear growth, BMI, body composition, spirometric lung volumes or self-assessed pubertal development, either overall or in subgroups defined by sex, calcium intake or baseline vitamin D status.

Null findings from the current study contrast with those from observational studies reporting independent associations between vitamin D deficiency and reduced lean mass,[8] slower linear growth,[7] childhood obesity[9–11] and precocious puberty.[12] However, they are consistent with those of the only other phase 3 RCT to investigate the effects of vitamin D on growth and development in school-age children that we are aware of, which reported no effect of weekly vitamin D supplementation on growth or development among Mongolian schoolchildren aged 6–13 years at baseline.[17] Contrasting findings from observational studies vs clinical trials may reflect the fact that the former are more susceptible to the effects of confounding or reverse causality. Although we observed limited evidence (P for interaction 0.049) to support the hypothesis that the effects of vitamin D on FVC might be modified by sex, this finding may have arisen because of type 1 error, given the multiplicity of analyses conducted.

Our study has several strengths. The intervention was sustained (3 years), allowing ample time for any effects of vitamin D supplementation on outcomes of interest to manifest. Moreover, the intervention was effective in elevating serum $25(OH)D$ concentrations, reflecting adequacy of the dosing regimen employed as well as good adherence resulting from directly observed administration of weekly supplements in schools during term time. Our large sample size and low rates of loss to follow-up maximised the power to detect modest effects of the intervention, particularly for outcomes that were assessed in the main trial population. Furthermore, we assessed a broad range of anthropometric and developmental outcomes, that included the use of DXA, the gold standard investigation for the assessment of body composition.

Our study also has some limitations. The baseline prevalence of vitamin D deficiency was low, perhaps reflecting plentiful exposure to sunshine in the study setting.[25] Dietary factors are less likely to contribute to this phenomenon, as frequency of intake of oily fish and

**Table 2** Mean height-for-age z-scores at annual follow-up by allocation, main study participants: overall and by subgroup

| | | Follow-up time point | Vitamin D arm: mean value (SD) (n) | Placebo arm: mean value (SD) (n) | Adjusted mean difference (95% CI)* | P for time point* | Overall P * | P for interaction* |
|---|---|---|---|---|---|---|---|---|
| Overall | | 1 year | −0.22 (1.12) (664) | −0.15 (1.07) (661) | −0.06 (−0.17 to 0.06) | 0.33 | 0.22 | – |
| | | 2 years | −0.14 (1.05) (613) | −0.06 (1.01) (606) | −0.04 (−0.16 to 0.07) | 0.46 | | |
| | | 3 years | −0.26 (1.04) (670) | −0.14 (1.05) (691) | −0.08 (−0.19 to 0.03) | 0.17 | | |
| By sex | Boys | 1 year | −0.31 (1.14) (304) | −0.27 (1.06) (306) | −0.02 (−0.17 to 0.14) | 0.85 | 0.42 | 0.94 |
| | | 2 years | −0.21 (1.11) (285) | −0.19 (0.95) (283) | 0.04 (−0.13 to 0.20) | 0.67 | | |
| | | 3 years | −0.34 (1.07) (310) | −0.28 (1.00) (326) | −0.03 (−0.19 to 0.13) | 0.69 | | |
| | Girls | 1 year | −0.15 (1.09) (360) | −0.04 (1.07) (355) | −0.10 (−0.25 to 0.06) | 0.21 | 0.34 | |
| | | 2 years | −0.08 (0.99) (328) | 0.05 (1.06) (323) | −0.12 (−0.28 to 0.04) | 0.14 | | |
| | | 3 years | −0.19 (1.02) (360) | −0.02 (1.08) (365) | −0.13 (−0.28 to 0.03) | 0.11 | | |
| By calcium intake† | <median | 1 year | −0.16 (1.05) (314) | −0.10 (1.02) (331) | −0.04 (−0.18 to 0.11) | 0.64 | 0.02 | 0.07 |
| | | 2 years | −0.17 (0.96) (288) | −0.05 (0.96) (310) | −0.04 (−0.19 to 0.11) | 0.64 | | |
| | | 3 years | −0.28 (0.99) (312) | −0.15 (0.99) (362) | −0.08 (−0.23 to 0.07) | 0.29 | | |
| | ≥median | 1 year | −0.25 (1.16) (327) | −0.18 (1.13) (314) | −0.05 (−0.21 to 0.12) | 0.60 | 0.72 | |
| | | 2 years | −0.07 (1.12) (307) | −0.06 (1.06) (279) | −0.03 (−0.20 to 0.14) | 0.76 | | |
| | | 3 years | −0.20 (1.09) (337) | −0.13 (1.11) (311) | −0.06 (−0.22 to 0.11) | 0.50 | | |
| By baseline 25(OH) D concentration‡ | <75 nmol/L | 1 year | −0.23 (1.13) (349) | −0.03 (1.11) (332) | −0.17 (−0.33 to −0.01) | 0.04 | 0.53 | 0.50 |
| | | 2 years | −0.14 (1.11) (325) | 0.05 (1.06) (323) | −0.12 (−0.28 to 0.04) | 0.15 | | |
| | | 3 years | −0.25 (1.10) (340) | −0.06 (1.07) (359) | −0.15 (−0.31 to 0.01) | 0.07 | | |
| | ≥75 nmol/L | 1 year | −0.22 (1.15) (187) | −0.25 (0.98) (201) | 0.02 (−0.17 to 0.22) | 0.82 | 0.15 | |
| | | 2 years | −0.17 (1.00) (171) | −0.08 (0.96) (179) | −0.07 (−0.26 to 0.13) | 0.50 | | |
| | | 3 years | −0.30 (1.00) (196) | −0.18 (0.99) (206) | −0.06 (−0.25 to 0.13) | 0.53 | | |

*Effect estimates and p values from mixed effects linear regression models including a fixed effect for allocation to vitamin D versus placebo, a random effect for allocation to vitamin D versus placebo, a random effect for repeated assessments of each individual participant and a random effect of school of attendance.
†Median calcium intake 466 mg/day.
‡Deseasonalised values.
.n, number; 25(OH)D, 25-hydroxyvitamin D.;

**Table 3** Mean BMI-for-age z-scores at annual follow-up by allocation, main study participants: overall and by subgroup

| | | | Vitamin D arm: mean value (SD)(n) | Placebo arm: mean value (SD) (n) | Adjusted mean difference (95% CI) | P for time point | Overall P | P for interaction |
|---|---|---|---|---|---|---|---|---|
| Overall | | 1 year | 0.16 (1.03) (664) | 0.17 (1.02) (661) | −0.02 (−0.14 to 0.09) | 0.68 | 0.86 | -- |
| | | 2 years | 0.15 (1.41) (613) | 0.25 (1.21) (606) | −0.09 (−0.20 to 0.03) | 0.14 | | |
| | | 3 years | 0.17 (1.07) (669) | 0.21 (1.17) (691) | −0.04 (−0.16 to 0.07) | 0.45 | | |
| By sex | Boys | 1 year | 0.08 (0.99) (304) | −0.00 (0.99) (306) | 0.09 (−0.08 to 0.27) | 0.30 | 0.59 | 0.29 |
| | | 2 years | 0.00 (1.68) (285) | −0.04 (1.30) (283) | 0.04 (−0.14 to 0.22) | 0.64 | | |
| | | 3 years | −0.02 (1.04) (309) | −0.12 (1.20) (326) | 0.09 (−0.09 to 0.26) | 0.33 | | |
| | Girls | 1 year | 0.23 (1.06) (360) | 0.32 (1.02) (355) | −0.14 (−0.28 to 0.00) | 0.06 | 0.26 | |
| | | 2 years | 0.28 (1.10) (328) | 0.50 (1.06) (323) | −0.21 (−0.36 to −0.06) | 0.00 | | |
| | | 3 years | 0.33 (1.06) (360) | 0.50 (1.07) (365) | −0.17 (−0.32 to −0.03) | 0.02 | | |
| By calcium intake[†] | <median | 1 year | 0.15 (1.01) (314) | 0.15 (1.03) (331) | −0.01 (−0.17 to 0.14) | 0.86 | 0.16 | 0.08 |
| | | 2 years | 0.18 (1.08) (288) | 0.21 (1.34) (310) | −0.03 (−0.19 to 0.13) | 0.75 | | |
| | | 3 years | 0.21 (1.07) (312) | 0.20 (1.14) (362) | 0.01 (−0.14 to 0.17) | 0.87 | | |
| | ≥median | 1 year | 0.18 (1.04) (327) | 0.22 (1.01) (314) | −0.04 (−0.21 to 0.13) | 0.61 | 0.27 | |
| | | 2 years | 0.13 (1.67) (307) | 0.31 (1.07) (279) | −0.15 (−0.32 to 0.03) | 0.10 | | |
| | | 3 years | 0.15 (1.07)(336) | 0.22 (1.23) (311) | −0.09 (−0.26 to 0.08) | 0.31 | | |
| By baseline 25(OH) D concentration[‡] | <75 nmol/L | 1 year | 0.30 (1.03) (349) | 0.28 (0.98) (332) | 0.03 (−0.14 to 0.19) | 0.76 | 0.31 | 0.31 |
| | | 2 years | 0.22 (1.64) (325) | 0.31 (1.33) (323) | −0.07 (−0.24 to 0.10) | 0.41 | | |
| | | 3 years | 0.25 (1.07) (340) | 0.34 (1.08) (359) | −0.09 (−0.25 to 0.08) | 0.31 | | |
| | ≥75 nmol/L | 1 year | 0.02 (0.97) (187) | 0.09 (1.02) (201) | −0.06 (−0.25 to 0.13) | 0.57 | 0.58 | |
| | | 2 years | 0.01 (1.05) (171) | 0.18 (1.00) (179) | −0.11 (−0.30 to 0.09) | 0.28 | | |
| | | 3 years | 0.09 (1.00) (196) | 0.06 (1.17) (206) | 0.02 (−0.17 to 0.21) | 0.87 | | |

*Effect estimates and p values from mixed effects linear regression models including a fixed effect for allocation to vitamin D versus placebo, a random effect for repeated assessments of each individual participant and a random effect of school of attendance.
†Median calcium intake 466 mg/day.
‡Deseasonalised values.
.n, number; 25(OH)D, 25-hydroxyvitamin D.;

**Table 4** End-study pubertal development indices by allocation, substudy participants: overall and by subgroup

| | | | Vitamin D arm: mean value (SD) (n) | Placebo arm: mean value (SD) (n) | Adjusted mean difference (95% CI) | P value | P for interaction |
|---|---|---|---|---|---|---|---|
| Males | Pubic hair, mean Tanner score (SD) (n) | Overall | 3.15 (0.91) (96) | 2.81 (0.99) (90) | 0.25 (−0.02 to 0.51) | 0.07 | – |
| | | Calcium intake† <median | 3.13 (0.93) (46) | 2.73 (1.14) (52) | 0.29 (−0.10 to 0.67) | 0.14 | 0.39 |
| | | ≥median | 3.15 (0.90) (48) | 2.92 (0.75) (38) | 0.16 (−0.19 to 0.52) | 0.37 | |
| | | Baseline 25(OH)D concentration‡ <75 nmol/L | 3.11 (0.92) (44) | 2.88 (0.92) (42) | 0.07 (−0.28 to 0.41) | 0.70 | 0.39 |
| | | ≥75 nmol/L | 3.18 (0.94) (34) | 2.70 (1.18) (30) | 0.43 (−0.07 to 0.93) | 0.09 | |
| | External genitalia, mean Tanner score (SD) (n) | Overall | 3.19 (0.91) (96) | 2.90 (0.99) (90) | 0.17 (−0.09 to 0.43) | 0.19 | – |
| | | Calcium intake† <median | 3.28 (0.89) (46) | 2.81 (1.10) (52) | 0.35 (−0.01 to 0.71) | 0.06 | 0.07 |
| | | ≥median | 3.10 (0.93) (48) | 3.03 (0.82) (38) | −0.04 (−0.40 to 0.33) | 0.84 | |
| | | Baseline 25(OH)D concentration‡ <75 nmol/L | 3.27 (0.82) (44) | 2.90 (0.91) (42) | 0.22 (−0.10 to 0.54) | 0.18 | 0.81 |
| | | ≥75 nmol/L | 3.15 (1.02) (34) | 2.87 (1.17) (30) | 0.30 (−0.17 to 0.77) | 0.21 | |
| Females | Pubic hair, mean Tanner score (SD) (n) | Overall | 3.10 (0.98) (105) | 3.11 (0.98) (99) | −0.09 (−0.35 to 0.17) | 0.50 | – |
| | | Calcium intake† <median | 3.15 (0.98) (47) | 3.15 (1.01) (46) | 0.01 (−0.38 to 0.40) | 0.96 | 0.60 |
| | | ≥median | 3.11 (0.92) (54) | 3.10 (0.97) (50) | −0.23 (−0.56 to 0.10) | 0.17 | |
| | | Baseline 25(OH)D concentration‡ <75 nmol/L | 3.14 (0.90) (59) | 3.13 (0.88) (55) | −0.03 (−0.35 to 0.28) | 0.83 | 0.94 |
| | | ≥75 nmol/L | 2.96 (1.22) (26) | 3.04 (1.08) (24) | −0.14 (−0.74 to 0.46) | 0.65 | |
| | Breast development, mean Tanner score (SD)(n) | Overall | 3.15 (0.87) (105) | 3.10 (0.91) (99) | −0.03 (−0.26 to 0.20) | 0.80 | – |
| | | Calcium intake† <median | 3.15 (0.91) (47) | 3.22 (0.92) (46) | −0.06 (−0.41 to 0.29) | 0.74 | 0.93 |
| | | ≥median | 3.20 (0.81) (54) | 3.04 (0.90) (50) | −0.05 (−0.36 to 0.26) | 0.76 | |
| | | Baseline 25(OH)D concentration‡ <75 nmol/L | 3.27 (0.85) (59) | 3.07 (0.86) (55) | 0.15 (−0.16 to 0.45) | 0.34 | 0.48 |
| | | ≥75 nmol/L | 3.08 (0.93) (26) | 3.08 (0.97) (24) | −0.08 (−0.55 to 0.38) | 0.73 | |
| | Proportion menstruating | Overall | 67/105 (63.81) | 60/99 (60.61) | 0.95 (0.51 to 1.77) | 0.86 | -- |
| | | Calcium intake† <median | 28/47 (59.57) | 26/46 (56.52) | 1.33 (0.50 to 3.56) | 0.57 | 0.24 |
| | | ≥median | 38/54 (70.37) | 34/50 (68.00) | 0.56 (0.21 to 1.52) | 0.26 | |
| | | Baseline 25(OH)D concentration‡ <75 nmol/L | 43/59 (72.88) | 35/55 (63.64) | 1.39 (0.59 to 3.28) | 0.45 | 0.59 |
| | | ≥75 nmol/L | 16/26 (61.54) | 15/24 (62.50) | 0.90 (0.26 to 3.15) | 0.87 | |
| | Mean age at menarche, years (SD) (n) | Overall | 11.94 (0.93) (66) | 11.86 (0.66) (58) | −0.03 (−0.31 to 0.25) | 0.84 | – |
| | | Calcium intake† <median | 11.89 (0.83) (28) | 11.71 (0.69) (24) | 0.12 (−0.26 to 0.49) | 0.54 | 0.32 |
| | | ≥median | 11.97 (1.01) (37) | 11.97 (0.63) (34) | −0.11 (−0.51 to 0.29) | 0.58 | |
| | | Baseline 25(OH)D concentration‡ <75 nmol/L | 12.00 (0.88) (42) | 11.88 (0.69) (34) | 0.04 (−0.31 to 0.39) | 0.82 | 0.64 |
| | | ≥75 nmol/L | 11.88 (1.15) (16) | 11.93 (0.62) (14) | −0.14 (−0.73 to 0.44) | 0.63 | |

*Effect estimates and p values from multilevel mixed models including a random effect for school of attendance.
†Median calcium intake 466 mg/day.
‡Deseasonalised values.
.n, number; 25(OH)D, 25-hydroxyvitamin D;.

other foods containing vitamin D did not associate with baseline vitamin D status in the study population.[25] Our results cannot therefore be generalised to settings where vitamin D deficiency is common. However, we highlight that results from our trial in Mongolia (where the baseline prevalence of vitamin D deficiency was much higher than we observed in the current study) were also null for outcomes of linear growth and body mass index.[17] Accordingly, null results from the current trial cannot be explained simply by the relatively high baseline vitamin D status of participants in Cape Town. A second potential limitation relates to the fact that pubertal status was assessed by participants themselves, rather than by health professionals, whose judgement may be more objective. However, the double-blind placebo-controlled trial design of our study will have distributed any resultant imprecision equally between study arms, ensuring that bias was not introduced. Moreover, pubertal self-assessment has previously been done by adolescent participants in another South African cohort study, and shown to be reliable.[27]

In conclusion, we report that oral vitamin D supplementation at a dose of 10 000 IU/week for 3 years was effective in elevating serum 25(OH)D concentrations in schoolchildren of black African ancestry living in Cape Town, South Africa. However, this was not associated with any effects on linear growth, body habitus, spirometric outcomes or pubertal development. Taken together with null results from a trial of weekly vitamin D supplementation with similar outcomes conducted in Mongolia,[17] our findings do not support the use of vitamin D supplements to influence child growth or development in populations where rickets have been excluded. Further research could be conducted to re-evaluate growth and developmental outcomes in the future, to determine whether any adverse effects might be associated with withdrawing vitamin D supplementation from individuals randomised to the intervention arm of this study, who may have become habituated to it during their participation in the trial.

**Author affiliations**
[1]Desmond Tutu HIV Centre, Institute of Infectious Disease & Molecular Medicine, University of Cape Town, Cape Town, South Africa
[2]Department of Medicine, University of Cape Town, Cape Town, South Africa
[3]Health through Physical Activity, Lifestyle and Sport Research Centre (HPALS), Department of Human Biology, Faculty of Health Sciences University of Cape Town, Cape Town, South Africa
[4]SAMRC/Wits Developmental Pathways for Health Research Unit, Department of Paediatrics, Faculty of Health Sciences University of the Witwatersrand, Johannesburg, South Africa
[5]Wolfson Institute of Population Health, Faculty of Medicine and Dentistry, Queen Mary University of London, London, UK
[6]Blizard Institiute, Faculty of Medicine and Dentistry, Queen Mary University of London, London, UK
[7]Centre for Infectious Diseases Research in Africa, Institute of Infectious Disease & Molecular Medicine, Faculty of Health Sciences University of Cape Town, Cape Town, South Africa
[8]Walter and Eliza Hall Institute of Medical Research, Melbourne, Victoria, Australia
[9]Department of Paediatrics and Child Health, University of Cape Town, Cape Town, South Africa
[10]Norwich Medical School, University of East Anglia, Norwich, UK
[11]Department of Laboratory Medicine, Norfolk and Norwich University Hospital NHS Foundation Trust, Norwich, UK
[12]MRC Lifecourse Epidemiology Centre, University of Southampton, Southampton, UK
[13]University Hospital Southampton NHS Foundation Trust, Southampton, UK
[14]The Francis Crick Institute, London, UK
[15]Department of Infectious Diseases, Imperial College London, London, UK

**Acknowledgements** We thank all the children who participated in the trial, and their parents/guardians; Professor Richard L Hooper for advice on statistical analysis; members of the Independent Data Monitoring Committee (Professor Guy Thwaites, Oxford University Clinical Research Unit, Ho Chi Minh City, Vietnam (Chair); Professor John Pettifor, University of the Witwatersrand, Johannesburg, South Africa; and Professor Sarah Walker, MRC Clinical Trials Unit, London, UK); and members of the Trial Steering Committee (Professor Beate Kampmann, London School of Hygiene and Tropical Medicine, London, UK (Chair); Professor Ashraf Coovadia, University of the Witwatersrand, Johannesburg, South Africa; Dr Karen Jennings, City Health, Cape Town, South Africa; Dr Robin Dyers, Department of Health, Western Cape Government, Cape Town, South Africa; and Dr Guy de Bruyn, Sanofi Pasteur, Swiftwater, Pennsylvania, USA). For the purposes of open access the author has applied a CC-BY public copyright to any author-accepted manuscript arising from this submission.

**Contributors** ARM conceived the study. KM, LM, AKC, JN, CC, NCH, RLH, RJW, L-GB and ARM contributed to study design and protocol development. KM led on trial implementation, with support from JS, DAJ, JN, L-GB and ARM. LM and AEM oversaw the performance of DXA scans. JT and WDF performed and supervised the conduct of biochemical assays. NW and AM drafted the statistical analysis plan. DAJ, KM, JS, NW and WM managed data. NW accessed, verified and analysed the data underlying the study. ARM and DAJ wrote the first draft of the trial report. All authors made substantive comments thereon and approved the final version for submission. ARM is guarantor for this report.

**Funding** This research was funded by the UK Medical Research Council (refs MR/R023050/1 and MR/M026639/1, both awarded to AM). RJW is supported by The Francis Crick Institute which receives funding from Wellcome (CC2112), Cancer Research UK (CC2112) and UK Research and Innovation (Medical Research Council CC2112). He also receives support from Wellcome (203135) and in part by the NIHR Biomedical Research Centre of Imperial College NHS Trust.

**Competing interests** None.

**Patient and public involvement** Patients and/or the public were involved in the design, or conduct, or reporting, or dissemination plans of this research. Refer to the Methods section for further details.

**Patient consent for publication** Not applicable.

**Ethics approval** The trial was approved by the University of Cape Town Faculty of Health Sciences Human Research Ethics Committee (Ref: 796/2015) and the London School of Hygiene and Tropical Medicine Observational/Interventions Research Ethics Committee (Ref: 7450-2). Participants gave informed consent to participate in the study before taking part.

**Provenance and peer review** Not commissioned; externally peer reviewed.

**Data availability statement** Data are available upon reasonable request. Anonymised data are available from the corresponding author on reasonable request, subject to terms of ethical and regulatory approvals.

**ORCID iD**
Adrian R Martineau http://orcid.org/0000-0001-5387-1721

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
