## [Reviewer comments · BMJ Paediatrics Open]

ARTICLE DETAILS

TITLE (PROVISIONAL)	Influence of vitamin D supplementation on growth, body composition, pubertal development and spirometry in South African schoolchildren: a randomised controlled trial (ViDiKids)
AUTHORS	Middelkoop, Keren Micklesfield, Lisa Stewart, Justine Walker, Neil Jolliffe, David A Mendham, Amy E Coussens, Anna K Nuttall, James Momand, Waheedullah Wilkinson, Robert J Bekker, Linda-Gail Tang, Jonathan Fraser, William cooper, cyrus Harvey, Nick Martineau, Adrian

VERSION 1 – REVIEW

REVIEWER	Dr. Peter Flom Peter Flom Consulting
REVIEW RETURNED	06-Feb-2024

GENERAL COMMENTS	I confine my remarks to statistical and methodological aspects of this paper. The general approach is fine, but I do have some issues to resolve before I can recommend publication. On p. 9, line 3-4: How does this inclusion criterion change the earlier ones? Do you mean that only children in grade 4 were eligible for this study? (It sounds like it, but I wasn't completely clear). p. 11 - sample size. "To detect an effect" is not specific effect. What size of effect? o lines 32 = 33 Why do subgroup analysis instead of looking at interactions? It's not wrong to do subgroup analysis, but interactions let you get a p value and CI for the effect.
---

	Categorizing continuous variables (e.g. initial vitamin D level, calcium intake) is generally a bad idea. It increases type I and type 2 error and imagines that something special happens right at the cutpoint. It would be better to leave these continuous and use splines to look for nonlinearities. p. 12 Why compare the substudy participants to the whole study? It's not wrong to do so, but I'm not sure it's necessary. Throughout the outcome section, rather than "did not affect" (or similar) put "did not significantly affect". p. 13 Here it appears you did do interactions rather than subgroup analysis. I commend you for not putting p values in Table 1 Tables that do include p values should have footnotes saying what test was used to get the p value. Table 2 It looks like the placebo arm did slightly better than the vit D arm. Peter Flom
--	--

REVIEWER	Dr. Harshpal Singh Sachdev Sitaram Bhatta Institute of Science and research, Pediatrics
REVIEW RETURNED	09-Feb-2024

GENERAL COMMENTS	This is a useful secondary analysis on a topical subject for which scant evidence is available. The primary study was conducted in an excellent manner and his prespecified analysis has also been conducted in a robust manner. The authors should be complimented for conducting this study and also taking pains to publish negative findings. They may wish to consider the following comments for potential modifications.  1. What was the rationale for using the Oral Vitamin D3 dose of 10,000 IU/week considering the Recommended Dietary Allowance for this age group? Was the intent to determine a physiological or a pharmacological effect? 2. The low baseline prevalence of Vitamin D deficiency needs to be explored a bit more. The study probably has data on food intakes since calcium intakes have been presented. Similarly, is it possible to quantify the dietary Vitamin D intakes? Is there a national or a school programme through which the children have access to Vitamin D fortified foods? 3. It would add value to include the bone density parameters from the available DXA data. 4. The manuscript looks somewhat incomplete without stating potential implications for policy and practice, and important future
---

	research possibilities. For the latter, would re-evaluation of the outcomes after some time be of interest to exclude the potential of harm of withdrawing Vitamin D from individuals who have got used to a higher dose for three years? Prof. Harshpal Singh Sachdev, FRCPCH Senior Consultant Pediatrics and Clinical Epidemiology, Sitaram Bhartia Institute of Science and Research, New Delhi 110016, India. Email: hpssachdev@gmail.com
--	--

VERSION 1 – AUTHOR RESPONSE

Reviewer 1, Comment 1: On p. 9, line 3-4: How does this inclusion criterion change the earlier ones? Do you mean that only children in grade 4 were eligible for this study? (It sounds like it, but I wasn't completely clear).

Response: The reviewer is correct: only children in grade 4 were eligible for the sub-study. We have added text below to clarify this point (new text underlined):

“An additional inclusion criterion for the sub-study was enrolment in Grade 4 at a participating school (i.e. only children in grade 4 were eligible for the sub-study).”

Reviewer 1, Comment 2: p. 11 - sample size. "To detect an effect" is not specific effect. What size of effect?

Response: We have revised the Methods to specify the effect sizes we were powered to detect in the main trial and the sub-study – new text underlined:

“Sample size for the main trial was calculated as the number needed to detect a 25% reduction in the proportion of children with a positive QFT-Plus assay result at 3-year follow-up (primary outcome) with 80% power and 5% type 1 error, assuming a 3.5% annual risk of QFT-Plus conversion, 20% loss to follow-up, and a 5% risk of an indeterminate QFT-Plus assay result at the end of the study [1]. Sample size for the sub-study was calculated as the number needed detect an inter-arm difference of 0.35 standard deviations in mean bone mineral content at the whole body less head and lumbar spine with 88% power and 5% type 1 error, assuming 29% loss to follow-up at 3 years [2].”

Reviewer 1, Comment 3: lines 32 = 33 Why do subgroup analysis instead of looking at interactions? It's not wrong to do subgroup analysis, but interactions let you get a p value and CI for the effect.

Response: Interaction analyses were done for all sub-group analyses. Text of Methods has been revised to clarify this point (new text underlined):

“Pre-specified sub-group analyses were conducted to determine whether the effect of vitamin D supplementation was modified by sex (male vs. female), baseline deseasonalised 25(OH)D3 concentration (<75 vs. ≥75 nmol/L)[3] and calcium intake (< vs. ≥ median value of 466 mg/day).[2] These were performed by repeating efficacy analyses with the inclusion of an interaction term between allocation (to vitamin D vs. placebo) and each posited effect-modifier with presentation of the P-value associated with this interaction term.”

Reviewer 1, Comment 4: Categorizing continuous variables (e.g. initial vitamin D level, calcium intake) is generally a bad idea. It increases type I and type 2 error and imagines that something special happens right at the cutpoint. It would be better to leave these continuous and use splines to look for nonlinearities.

Response: Our statistical analysis plan pre-specified sub-group analyses using cut-points for continuous variables as per the Tables. These cut-points were selected on a biological basis, reflecting the hypothesis that benefits of the intervention were more likely to be seen for example in people who were vitamin D insufficient vs. sufficient at baseline. Also, our experience is that readers who do not have advanced training in statistics find it easiest to understand /interpret presentation of interaction analyses when effects of the intervention are presented for specific sub-groups, differentiated from each other using a clinically meaningful cut-off. If we were to modify the paper to present analyses treating putative effect-modifiers as continuous variables, we are concerned that this could introduce type 1 error, since we would be diverging from our pre-specified plan, and introducing an additional exploratory analysis. Moreover, if this new approach yielded P values for interaction <0.05, readers would justifiably question why we diverged from our pre-specified plan, and could accuse us of modifying this plan in order to attain statistically significant results. Thus, for all these reasons, we prefer to retain our current approach.

Reviewer 1, Comment 5: p. 12 Why compare the substudy participants to the whole study? It's not wrong to do so, but I'm not sure it's necessary.

Response: We don't make formal statistical comparisons between these groups. However, we feel that it is important to highlight to the reader that sub-study participants (in whom body composition / developmental outcomes were assessed) formed a distinct (i.e. older) sub-set within the study population as a whole (in whom growth outcomes were assessed). It is important for the reader to be clear on this point when interpreting our findings.

Reviewer 1, Comment 6: Throughout the outcome section, rather than "did not affect" (or similar) put "did not significantly affect".

Response: requested changes made.

Reviewer 1, Comment 7: p. 13 Here it appears you did do interactions rather than subgroup analysis.

Response: As per our response to Comment 3 from this reviewer, interaction analyses were performed for all sub-group effects tested. Methods have been revised to clarify this point.

Reviewer 1, Comment 8: Tables that do include p values should have footnotes saying what test was used to get the p value.

Response: Footnotes added to all tables including p values.

Reviewer 1, Comment 9: Table 2 It looks like the placebo arm did slightly better than the vit D arm.

Response: P values in Table 2 are all >0.05 – to minimize risk of type 1 error we prefer not to speculate re any difference in outcomes between arms here.

Reviewer 2, Comment 1: What was the rationale for using the Oral Vitamin D3 dose of 10,000 IU/week considering the Recommended Dietary Allowance for this age group? Was the intent to determine a physiological or a pharmacological effect?

Response: Our intent was to determine a physiological effect. A weekly dose of 10,000 IU (equivalent to 1,429 IU/day) was selected in preference to a daily dose of 600 IU (the RDA for this age group) because we were concerned that the latter dose would be inadequate to maintain serum 25(OH)D concentrations >50 nmol/L, [4] and we felt that adherence to a directly-observed weekly dose would be superior to daily self-administration [5]. Weekly supplementation has also been shown to be effective in elevating 25(OH)D concentrations into the physiological range in children by other investigators [6]. Our approach appears justified by our findings that this dose did not elevated 25(OH)D concentrations into the supraphysiological range in any participant.

We have added the following text to Methods to justify our choice of dose:

“A weekly dose of 10,000 IU (equivalent to 1,429 IU/day) was selected in preference to a daily dose of 600 IU (the RDA for this age group) because we were concerned that the latter dose would be inadequate to maintain serum 25(OH)D concentrations >50 nmol/L,[4] and we felt that adherence to a directly-observed weekly dose would be superior to daily self-administration.[5] Weekly supplementation has also been shown to be effective in elevating 25(OH)D concentrations into the physiological range in children by other investigators.[6]”

Reviewer 2, Comment 2: The low baseline prevalence of Vitamin D deficiency needs to be explored a bit more. The study probably has data on food intakes since calcium intakes have been presented. Similarly, is it possible to quantify the dietary Vitamin D intakes? Is there a national or a school programme through which the children have access to Vitamin D fortified foods?

Response: We did collect information on intake of foods containing vitamin D, but this was semi-quantitative only. Our analysis of determinants of baseline vitamin D status in this population found that season, age and BMI were the principal determinants; intake of oily fish and other foods containing vitamin D did not associate with children’s vitamin D status [7]. There is no national / school programme of vitamin D fortification in South Africa.

We have added the following text to the Discussion (new text underlined) to address the Reviewer’s point:

“Baseline prevalence of vitamin D deficiency was low, perhaps reflecting plentiful exposure to sunshine in the study setting [7]. Dietary factors are less likely to contribute to this phenomenon, as frequency of intake of oily fish and other foods containing vitamin D did not associate with baseline vitamin D status in the study population [7].”

Reviewer 2, Comment 3: It would add value to include the bone density parameters from the available DXA data.

Response: these outcomes have now been reported elsewhere. Text of the current manuscript has been amended to clarify this point (Methods, p9):

“DXA outcomes relating to effects of the trial intervention on bone mineral content have been reported elsewhere.[2]

Reviewer 2, Comment 4: The manuscript looks somewhat incomplete without stating potential implications for policy and practice, and important future research possibilities. For the latter, would re-evaluation of the outcomes after some time be of interest to exclude the potential of harm of withdrawing Vitamin D from individuals who have got used to a higher dose for three years?

Response: Thank you for this important suggestion. We have added the following text to the end of

the article to address implications for policy, practice, and future research:

“Taken together with null results from a trial of weekly vitamin D supplementation with similar outcomes conducted in Mongolia [8], our findings do not support the use of vitamin D supplements to influence child growth or development in populations where rickets has been excluded. Further research could be conducted to re-evaluate growth and developmental outcomes in the future, to determine whether any adverse effects might be associated with withdrawing vitamin D supplementation from individuals randomised to the intervention arm of this study, who may have become habituated to it during their participation in the trial.”

We hope the responses above, along with the associated edits to the manuscript, address all concerns raised, and look forward to your assessment in due course.

Yours sincerely

Adrian Martineau, FRSB
Professor of Respiratory Infection and Immunity

VERSION 2 – REVIEW

REVIEWER	Dr. Peter Flom Peter Flom Consulting
REVIEW RETURNED	28-Feb-2024
GENERAL COMMENTS	The authors have addressed my concerns and i now recommend publication.

VERSION 2 – AUTHOR RESPONSE

None